

# Can squirrel monkeys learn an $AB^nA$ grammar? A re-evaluation of Ravignani et al. (2013)

Stefano Ghirlanda

Department of Psychology, Brooklyn College, Brooklyn, NY, United States of America
Departments of Psychology and Biology, CUNY Graduate Center, New York, NY, United States of America
Centre for the Study of Cultural Evolution, Stockholm University, Stockholm, Sweden

## ABSTRACT

*Ravignani et al. (2013)* habituated squirrel monkeys to sound sequences conforming to an $AB^nA$ grammar ($n = 1, 2, 3$), then tested them for their reactions to novel grammatical and non-grammatical sequences. Although they conclude that the monkeys "consistently recognized and generalized the sequence $AB^nA$," I remark that this conclusion is not robust. The statistical significance of results depends on specific choices of data analysis, namely dichotomization of the response variable and omission of specific data points. Additionally, there is little evidence of generalization to novel patterns ($n = 4, 5$), which is important to conclude that the monkeys recognized the $AB^nA$ grammar beyond the habituation patterns. Lastly, many test sequences were perceptually similar to habituation sequences, raising the possibility that the monkeys may have generalized based on perceptual similarity rather than based on grammaticality.

# INTRODUCTION

Within a wider program of research aimed at charting the evolution of linguistic abilities, *Ravignani et al. (2013)* tested whether squirrel monkeys (*Saimiri sciureus*) can detect violations of an $AB^nA$ grammar. The grammar was instantiated using sequences of low and high pitch sounds. Grammatical sequences were composed of a variable number of sounds of similar pitch (the *B*'s), sandwiched between two sounds similar in pitch to each other (the *A*'s), but very different from the *B*'s (Table 1). The study follows the general design of habituation/dishabituation studies (also known as familiarization/discrimination), used with both human infants (*Eimas et al., 1971*) and non-human animals (*Cheney & Seyfarth, 1988*; *Fischer, 2006*). The authors first habituated the monkeys to sequences with structure *ABA*, *ABBA*, and *ABBBA*, by playing these sequences for a total of 360 times over two days. I indicate these sequences as $AB^{1,2,3}A$ sequences. Following habituation, the authors conducted two tests to ascertain whether the animals would:

1. Show habituation to *other* grammatical sequences.
2. Show lack of habituation to non-grammatical sequences, such as *BA* or *ABB*.

Corresponding author
Stefano Ghirlanda,
drghirlanda@gmail.com

**Table 1** **Characteristics of sounds used to assemble stimulus sequences.** "Interval" refers to the difference between adjacent sounds. "JND" refers to the just-noticeable difference in the considered frequency range, i.e., the difference that is detected with 50% probability. JND data from *Wienicke, Häusler & Jürgens (2001)*.

| Sound class | Number of sounds | Frequency range (Hz) | Interval (Hz) | JND (Hz) |
|---|---|---|---|---|
| A | 44 | 1,800–2,200 | 9 | $\sim 70$ |
| B | 44 | 9,000–11,000 | 45.5 | $\sim 50$ |

Such a pattern of behavior, if found, would indicate that the monkeys could generalize the $AB^nA$ structure heard during habituation to novel sequences. In Test 1, the animals listened to grammatical and non-grammatical sequences composed of the same sounds heard during habituation. In Test 2, the role of high and low pitch tones was reversed. That is, whereas $A$ and $B$ had signified, respectively, low and high pitch sounds prior to Test 2, the opposite was true in Test 2.

According to *Ravignani et al. (2013)*, both Test 1 and 2 indicated that squirrel monkeys can detect the presence or absence of $AB^nA$ structure in novel sequences, including longer sequences in Test 1 ($n = 4, 5$) and novel pitch patterns in Test 2. Here, I show that this claim rests on several details of the authors' analysis, which are necessary to obtain conventionally "significant" results ($p < 0.05$) in both Test 1 and Test 2. Namely, significance is typically attained only when using a dichotomized response measure and specific criteria of data selection, such as excluding responses to $AB$ and $BA$ sequences in Test 2. Moreover, in Test 1, there was no direct demonstration of generalization to longer sequences ($n = 4, 5$). Lastly, generalization may have been based, at least partly, on perceptual similarity rather than grammaticality, as some test sequences were perceptually very similar to habituation sequences.

## MATERIALS AND METHODS

I acquired the data posted alongside the original article (*Ravignani et al., 2013*) and recast them as a table with the format displayed in Table 2. All analyses were performed with R, version 3.3.3 (*R Core Team, 2017*). The R code and the reformatted data are available as Supplemental Information 1.

## RESULTS

### Replication of results in *Ravignani et al. (2013)*

To ascertain that data acquisition did not introduce errors, this section reproduces the main analyses in *Ravignani et al. (2013)*. It also serves to summarize the original results, in order to understand how they are affected by the factors discussed later. Responding to grammatical and non-grammatical sequences in Tests 1 and 2 is displayed in Table 3. Results for Test 1 match the height of the bars in Fig. 2 of *Ravignani et al. (2013)*. Results for non-grammatical sequences in Test 2, however, are lower than in the original figure. The reason is that original analysis excluded responses to sequences $AB$ and $BA$

**Table 2   A random sample of reformatted experimental data.** The Response column indicates how many times a subject was observed orienting toward a speaker that was playing the test sequence.

| Test | Subject | Sequence | Response | Grammatical |
|------|---------|----------|----------|-------------|
| 1 | Co | ABBBBBA | 1 | Yes |
| 1 | Mo | ABBB | 1 | No |
| 1 | Ch | BBBBA | 2 | No |
| 1 | Ti | BBBA | 1 | No |
| 1 | Mo | ABBA | 2 | Yes |
| 1 | An | ABBB | 2 | No |
| 2 | Ch | ABA | 0 | Yes |
| 2 | Pi | ABBBBBA | 0 | Yes |
| 2 | Pi | BBBA | 1 | No |
| 2 | Mo | BBBA | 2 | No |

**Table 3   Results of Tests 1 and 2, all data included.** The $n$ column indicates the number of stimulus presentations for each condition.

| Test | Grammatical | $n$ | Proportion of trials with > 0 responses |
|------|-------------|-----|------------------------------------------|
| 1 | No | 48 | 0.77 |
| 1 | Yes | 48 | 0.60 |
| 2 | No | 32 | 0.78 |
| 2 | Yes | 32 | 0.62 |

**Table 4   Results of Test 2 after exclusions of stimuli *AB* and *BA*.** The $n$ column indicates the number of stimulus presentations for each condition.

| Test | Grammatical | $n$ | Proportion of trials with > 0 responses |
|------|-------------|-----|------------------------------------------|
| 2 | No | 24 | 0.83 |
| 2 | Yes | 32 | 0.62 |

(see 'Omission of data from analysis' for further analysis of this point). Excluding these sequences, I obtain Table 4, whose content matches the height of the bars for Test 2 in the original Fig. 2.

I also reproduced the original statistical results. The authors performed two paired $t$-tests, comparing for each subject the proportion of grammatical and non-grammatical sequences to which at least one response was recorded. For these tests, my analysis yields Table 5, rows 1 and 9, which match the results by *Ravignani et al. (2013)* apart from these authors reporting $t = 4.64$ rather than $t = 4.63$ for Test 2.

Lastly, *Ravignani et al. (2013)* conducted an ANOVA of the whole data set, with proportion of trials with a response as dependent variable and grammaticality and test as independent variables. The results showed a significant effect of grammaticality, which is reproduced in Table 6. Responses to sequences *AB* and *BA* from both Test 1 and Test 2 were excluded from this analysis.

**Table 5  Results of *t*-tests discussed in the main text, sorted by test number and excluded sequences.**
Rows referring to the same test and same excluded sequences, but different dichotomization, are adjacent to show the effect of dichotomization. The *n* column indicates the number of stimulus presentations included in the test, across all subjects. The number of experimental subjects is *d.f.* +1. Row numbers are for reference. Statistical significance at the 0.05 level is indicated by ★.

| | Test | Excluded sequences | Dichotomization | *n* | *t* | *d.f.* | *p* |
|---|---|---|---|---|---|---|---|
| 1 | 1 | – | Yes | 96 | 3.16 | 5 | 0.025★ |
| 2 | | | No | 96 | 1.66 | 5 | 0.16 |
| 3 | 1 | $AB^{1,2,3}A$ | Yes | 72 | 0.955 | 5 | 0.38 |
| 4 | | | No | 72 | 2.71 | 5 | 0.042★ |
| 5 | 1 | $AB^{4,5}A$ | Yes | 72 | 3.84 | 5 | 0.012★ |
| 6 | | | No | 72 | 0.655 | 5 | 0.54 |
| 7 | 2 | – | Yes | 64 | 1.99 | 3 | 0.14 |
| 8 | | | No | 64 | 0.835 | 3 | 0.47 |
| 9 | 2 | *AB*, *BA* | Yes | 56 | 4.63 | 3 | 0.019★ |
| 10 | | | No | 56 | 1.01 | 3 | 0.39 |

**Table 6  Results of ANOVA for Tests 1 and 2 combined, with a dichotomized response variable and excluding sequences *AB* and *BA*.**  Test based on 140 stimulus presentations. Statistical significance at the 0.05 level is indicated by ★.

| | Sum of Squares | *d.f.* | *F* | *p* |
|---|---|---|---|---|
| Grammatical | 1.7 | 1 | 8.33 | 0.0045★ |
| Test | 0.0048 | 1 | 0.02 | 0.88 |
| Grammatical:Test | 0.0036 | 1 | 0.02 | 0.89 |
| Residuals | 27 | 136 | | |

## Dichotomization of the dependent variable

*Ravignani et al. (2013)* counted the number of times monkeys turned their head toward a speaker, within 7 s from stimulus onset. Before performing the analyses reproduced above, they dichotomized this measure so that 0 head turns was "no response" and ≥1 head turns was "response." Dichotomization may have advantages:

1. It may make analysis and reporting simpler.
2. It may make observations more robust to noise. An orientation response, for example, may be interrupted by an extraneous cause (e.g., something catching the animal's attention) and then resumed. This would count as two separate responses without dichotomization, but as one response with dichotomization.

Dichotomization, however, also has disadvantages:

1. It discards information in the data, which may cause false positives and false negatives (*MacCallum et al., 2002*; *DeCoster, Iselin & Gallucci, 2009*). For example, it may be argued that an animal that orients toward a stimulus multiple times shows more surprise than an animal that orients only once, but this information is lost with dichotomization. Similarly, an animal that resumes an interrupted orientation response (see point 2 above) may be considered to show more interest in the stimulus than an animal that does not.
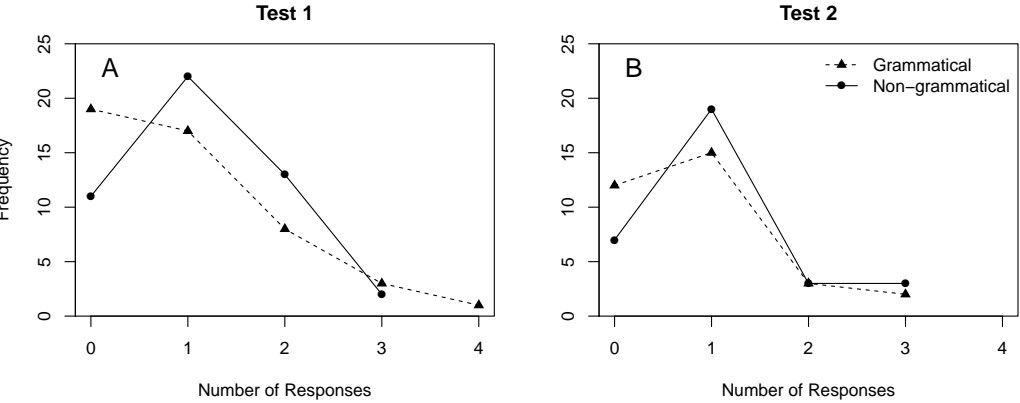

**Figure 1** Distribution of responses to grammatical and non-grammatical sequences in Tests 1 and 2.

**Table 7** Results of the ANOVA in Table 6 when responses are not dichotomized.

|  | Sum of Squares | d.f. | F | p |
|---|---|---|---|---|
| Grammatical | 1.7 | 1 | 2.21 | 0.14 |
| Test | 0.34 | 1 | 0.45 | 0.5 |
| Grammatical:Test | 0.008 | 1 | 0.01 | 0.92 |
| Residuals | 103 | 136 |  |  |

2. It may make observations more vulnerable to noise. For example, an animal may orient once toward a stimulus by chance, but orienting twice or more by chance is unlikely. If chance responses are sufficiently frequent, dichotomization may inflate the number of responses.

Statisticians generally advise against dichotomization (*MacCallum et al., 2002*; *DeCoster, Iselin & Gallucci, 2009*), but whether it is beneficial, neutral, or detrimental has not been studied in the case of habituation/dishabituation experiments. Here, it suffices to point out that results may differ depending on whether dichotomization is applied or not, in which case it may not be possible to reach firm conclusions.

*Ravignani et al. (2013)* observed multiple responses in 24% of test trials. Figure 1 shows the full distribution of responses in Test 1 and Test 2. Repeating the analyses in *Ravignani et al. (2013)* without dichotomization yields no significant difference between grammatical and non-grammatical sequences (Table 5, rows 2 and 10). Results for the ANOVA of the whole data set also become non-significant when responses are not dichotomized (compare Tables 6 and 7).

## Omission of data from analysis

*Ravignani et al. (2013)* excluded from the analysis of Test 2 results the responses elicited by two stimulus sequences, *AB* and *BA*. All analyses reported so far have honored this choice, but there are reasons to revisit it. The rationale for excluding responses to *AB* and *BA* was that this pair of sequences, being symmetrical with respect to the exchange of low and high pitch tones introduced in Test 2, had also been presented in Test 1. Thus,

the authors argued, the monkeys could have habituated to these sequences during Test 1, confounding the outcome of Test 2. The data, however, do not show any habituation: the monkeys responded equally to *AB* and *BA* in Test 1 and Test 2 (an average of 1 head turn per presentation). This is expected from the typical finding that habituation proceeds over many trials (*Bouton, 2016*), while *AB* and *BA* were presented only once to each animal in Test 1.

Without excluding *AB* and *BA*, even the original data analysis yields non-significant results in Test 2, as pointed out by *Ravignani et al. (2013)* themselves (Table 5, row 7). The analysis stays non-significant if responses are not dichotomized (Table 5, row 8).

## Generalization to $AB^{4,5}A$ sequences

Test 1 presented both sequences with the familiar $AB^{1,2,3}A$ structure (heard during habituation) and longer sequences with the novel structure $AB^{4,5}A$. These have different roles in assessing generalization: $AB^{1,2,3}A$ sequences probe generalization with respect to sound identity, while $AB^{4,5}A$ sequences probe generalization with respect to sequence length (*Ravignani et al., 2013*). The latter is theoretically important to claim full mastery of the $AB^{n}A$ grammar (*Fitch & Friederici, 2012*), although it is not always tested in practice (*Rey, Perruchet & Fagot, 2012*; *Perruchet & Rey, 2015*). *Ravignani et al. (2013)* did include longer sequences in their tests, but they analyzed responses to these sequence together with responses to $AB^{1,2,3}A$ sequences, and thus did not directly demonstrate generalization to novel sequence length. Unfortunately, small sample size makes it hard to assess generalization to $AB^{1,2,3}A$ and $AB^{4,5}A$ sequences separately. For completeness, Table 5 reports the results of *t*-tests after selectively excluding from analysis either $AB^{1,2,3}A$ sequences (rows 3 and 4) or $AB^{4,5}A$ sequences (rows 5 and 6), with or without dichotomization. As with the other tests reported above, the conventional significance of these tests changes with dichotomization, yielding no firm conclusion regarding generalization to $AB^{1,2,3}A$ and $AB^{4,5}$ sequences separately.

Test 2 suffers less from this confound because the pitch pattern of test sequences was different from that of habituation sequences. Thus, even $AB^{1,2,3}A$ sequences can be considered structurally novel. As detailed in section 'Omission of data from analysis', however, Test 2 also provides little evidence of generalization.

## Contribution of perceptual similarity

Attempts to reveal generalization based on abstract patterns must control for animals' tendency to generalize based on perceptual similarity. For example, it would not be surprising to find similar responses to two *ABA* sequences composed of almost identical *A* and *B* sounds, because animals typically respond in the same way to stimuli that are perceptually very close to each other (*Mackintosh, 1974*; *Ghirlanda & Enquist, 2003*). Generalization based on perceptual closeness operates across many animal taxa, including mammals, birds, fish, and insects, and does not require grammatical abilities (*Enquist & Ghirlanda, 2005*). *Ravignani et al. (2013)* addressed this concern by assembling stimulus sequences out of two sets of 44 sounds each (one for the *A*'s, one for the *B*'s), and by selecting the *A* and *B* sounds randomly when playing each sequence. In this way, the chance of a test

sequence being identical to a habituation sequence was small. Test sequences, however, may still have been perceptually close to habituation sequences. As summarized in Table 1, the 44 $A$ sounds spanned a range of about six JNDs, and could be as little as 0.15 JNDs apart (JND: just noticeable difference, a difference that is detected with 50% probability). $B$ sounds were about one JND apart. Data from stimulus generalization studies shows that generalization can be substantial over several JNDs (*Ghirlanda & Enquist, 2003*). During habituation, the monkeys heard 720 $A$ sounds and 1680 $B$ sounds, meaning that each $A$ and $B$ was likely heard in all sequence positions within the habituation set of $AB^{1,2,3}A$ sequences. These experiences may have been sufficient to induce generalization based on perceptual similarity.

## DISCUSSION

*Ravignani et al. (2013)* concluded that "Squirrel monkeys consistently recognized and generalized the sequence $AB^nA$" and that they "are sensitive to abstract dependencies of different lengths and can generalize to new lengths and auditory parameters of the stimuli" ("dependency" here indicates that grammatical sequences were bound to have identical first and last elements). I have argued that these conclusions should be tempered because of the following circumstances:

1. The conventional significance of test results changes depending on whether the response variable is dichotomized, and we do not know whether dichotomizing is appropriate ('Dichotomization of the dependent variable').
2. Including stimuli $AB$ and $BA$ in the analysis of Test 2 renders results non-significant (with or without dichotomization). The rational for not including these sequences is weak ('Omission of data from analysis').
3. The tests provided no evidence of generalization to sequences longer than the habituation sequences, which is necessary to claim recognition of the $AB^nA$ grammar ('Generalization to $AB^{4,5}A$ sequences').
4. Generalization based on perceptual similarity, rather than on pattern, may have contributed to performance in Test 1 ('Contribution of perceptual similarities').

I have reached these conclusions following the same data analysis strategy as *Ravignani et al. (2013)*, based on the $t$-tests detailed in 'Replication of results in *Ravignani et al. (2013)*' and presented in Table 5. An alternative strategy, based on logistic regression for dichotomized responses and on Poisson regression for non-dichotomized responses, leads to the same results (not reported).

Points 1 and 2 above are largely statistical, and indicate that the results are not very robust (by the definition of robust as "insensitive to changes in details"). To resolve these points, the best strategy would be to collect more data (although animals may eventually habituate to all test sequences), and to understand whether multiple response are informative about the underlying cognition, in order to decide whether dichotomization is appropriate. It may also be helpful to administer more habituation trials, to increase the potential difference between novel and familiar sequences.

Points 3 and 4 are methodological, and signal the need for a more interpretable testing strategy. For example, habituation and test sequences could be designed to decrease

perceptual similarity. The number of stimulus presentations could be increased to ensure sufficiently many responses to test generalization to $AB^{4,5}A$ sequences separately from generalization to $AB^{1,2,3}A$ sequences. It may also be informative to test non-grammatical sequences similar to $AB^{4,5}A$ sequences, such as $B^{4,5}A$ and $AB^{4,5}$.

## ACKNOWLEDGEMENTS

I am grateful to Andrea Ravignani and Tecumseh Fitch for discussion, and to two anonymous reviewers for feedback on previous versions.

### Funding

This work was partially supported by grant 2015-0005 from the Knut and Alice Wallenberg Foundation. There was no additional external funding received for this study. The funders had no role in study design, data collection and analysis, decision to publish, or preparation of the manuscript.

### Grant Disclosures

The following grant information was disclosed by the author:
Knut and Alice Wallenberg Foundation: 2015-0005.

### Competing Interests

The authors declare there are no competing interests.

### Author Contributions

- Stefano Ghirlanda conceived and designed the experiments, performed the experiments, analyzed the data, contributed reagents/materials/analysis tools, wrote the paper, prepared figures and/or tables, reviewed drafts of the paper.

### Data Availability

   The analysis code and raw data have been supplied as Supplemental Files.

### Supplemental Information

Supplemental information for this article can be found online at http://dx.doi.org/10.7717/peerj.3806#supplemental-information.

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
