# Peer review of "Can squirrel monkeys learn an ABnA grammar? A re-evaluation of Ravignani et al. (2013)"

_PeerJ, doi:10.7717/peerj.3806_

## Round 0.1 · original submission · Major Revisions

· Academic Editor

Major Revisions

I have been very fortunate to receive two very thoughtful and detailed reviews from experts in this field. I agree with the reviewers that the general aim of the paper is laudable and believe that publication of such a response is warranted. However, both reviewers have identified several quite distinct concerns with the current version of the MS, which I will not reiterate here. I am willing to invite a revision but you must address the concerns of the reviewers both in terms of your claims/recommendations and presentation of the previous literature.

Reviewer 1 ·

Basic reporting

Generally the paper is clearly written and well structured. A number of minor spelling mistakes and typos should be corrected before publication.

Experimental design

The article presents a reanalysis of existing data. The analyses are conducted and reported appropriately.

Validity of the findings

The key reanalyses are clear and accurate. I remain unconvinced that the figure of 6% of the variance explained is a fair and reasonable reflection of the effect size shown in the original study (see comments, below), but in general the analyses are correctly performed and their results appear valid.

Additional comments

Review of “Can squirrel monkeys learn an ABnA grammar? A re-evaluation of Ravignani et al. (2013)” Ghirlanda, S.

In this article, Ghirlanda summarizes two experiments by Ravignani et al. (2013), in which squirrel monkeys were habituated to sequences of the form ABnA, where A and B tokens represent high and low pitches. The monkeys were then presented with “non-grammatical” sequences that did not conform to the ABnA pattern. Ravignani et al., conclude that the results demonstrate that the monkeys learned the non-adjacent dependencies between the ‘A’ sounds, and generalized this learning to novel sequences and pitch classes. Ghirlanda replicates the analyses performed in the original study and raises concerns regarding some of the decisions taken by the original authors (dichotomization of the data; excluding responses to certain stimuli), and challenges their conclusions.

Comment 1
In general, I think that the issues raised in this article are reasonable and important, particularly regarding the statistical analyses of the data in Ravignani et al. Ghirlanda presents some important and valid criticisms, alongside some points which are more contentious and less clearly supported by the data. A key issue raised in this paper is the decision to dichotomize the data into sequences that received one or more responses and those that elicited zero responses. Ghirlanda demonstrates that if the data is instead considered based on the number of responses each sequence elicited, the reported effects become non-significant. I agree with this reanalysis, and think this is an important point that should be published so that the original results can be interpreted appropriately. However, I have a number of comments that should be addressed before I would recommend publication of the article.

Comment 2
Ghirlanda suggests that it would be prudent to exclude sequences with the structure AB1,2,3A from the analysis of experiment 1, as these were the structures that the monkeys were habituated to. However, I am not clear what stimuli would be left to analyze if these stimuli were removed from the dataset. Based on the sequences shown in Table 1, this would leave only a single grammatical sequence (ABBBBBA). The Ravignani paper suggests that the sequence ABBBBA was also presented during test 1. Removing all of these data points would severely limit the power of the analysis (potentially explaining the non-significant result obtained). Moreover, it is possible that the monkeys may respond strongly to the sequences ABBBBBA (and ABBBBA) simply because they are longer than anything that they were habituated to, and therefore might be likely to provoke responses.
The issue of whether or not the monkeys were able to generalize to new sequences or structures is an important one that should be raised, and the design of the Ravignani et al. experiment could certainly be challenged here. However, I am not convinced that simply omitting data and performing new analyses provides reasonable results. If a large portion of data were omitted from any experiment one would expect the results to change, and in the direction of becoming less significant. Ghirlanda raises a reasonable criticism, but this should inform stimulus design in future experiments, rather than form the basis for post-hoc reanalysis of existing data.

Comment 3
In the abstract, Ghirlanda states that pattern grammaticality can account for at most 6% of the variance in responding (based on an ANOVA, line 142). The author’s own calculation, that monkeys responded to 83% of non-grammatical sequences and 62% of grammatical ones, gives a similar value of 7% better than chance (83/(83+62) = 57%, chance = 50%). However, I am not convinced that this calculation appropriate reflects the effect reported in the original experiments. The value of 57% represents the probability that a response was elicited by non-grammatical sequence. This might be appropriate if the animals were trained to respond to non-grammatical sequences and to withhold response to grammatical ones (as in a go/no-go task). However I do not believe this calculation is appropriate in a natural response paradigm such as this one, and it could be quite misleading to readers. Using the response rates in the article, one could calculate that the monkeys responded 34% more frequently to the non-grammatical than the grammatical sequences (83/62 = 1.34). This represents a modest but noteworthy difference in responses.
This type of experiment asks whether animals are sensitive to the rules of a grammar, or the properties of the sequences. They do not require that the animals must respond to every non-grammatical sequence and never respond to a grammatical one. Why should the animals naturally respond in such a binary way? After all, even a grammatical sequence might be somewhat interesting to a monkey – but critically they appear to be less interesting than the non-grammatical sequences. Ravignani and colleagues never make the claim that monkeys orienting responses are solely dictated by the grammaticality of the sequences, only that grammaticality affects the animals’ natural responses to auditory stimuli. This calculation, particularly reported in the abstract without context, is unhelpful in interpreting the results of an experiment using a natural response paradigm.
I think it is reasonable to include these calculations in the manuscript, but I think the figure of 6% should be removed from the abstract, where there is insufficient space to explain or discuss it properly, as it may mislead readers regarding the effects observed.

Comment 3
The author makes the point that as well as appealing to formal language theory, this experiment could be viewed in the context of work on animal working memory. Specifically, Ghirlanda states:
“The current task is similar [to the successive match to sample paradigm], in that all habituated sequences could be identified by retaining in memory the first sound, and then checking whether the last was the same or not [1]. Intervening sounds could be ignored (although we may expect them to make the task harder by introducing interfering memories). From this perspective, the question is not whether squirrel monkeys can conceptualize grammars with long-range dependencies, but whether their working memory span is sufficient to compare the first and last stimuli in a sequence.”
I agree with the first part of this statement – it seems very likely that the animals were comparing the first and last elements of the sequences. However, I think the last sentence suggests a possible misunderstanding of the goal of this type of research. Yes, this task requires that the working memory span of the squirrel monkey is long enough to compare the first and the last stimuli in a sequence. However, the discrimination of non-grammatical and grammatical stimuli requires that the animals implicitly learned that the relationships between the first and the last element was an important property of the stimuli in the first place. It is not only a question of whether they can hold the first element in memory, the key implication of Ravignani et al is that the animals were able to learn that sequences must begin and end with the same category of sound, and they responded differently when this does not occur. In this case, how they do so (hold first element in memory) is secondary to the fact that they learned to do so at all, which requires recognizing this pattern from the habituation sequences.
I wonder if this confusion is related to the fact that the author repeatedly refers to the habituation/dishabituation paradigm used by Ravignani et al., as a “task”. Unlike the match to sample tasks the author cites, the squirrel monkeys were not required to make any sort of responses. There was no task, per se, just an opportunity to make (or not make) natural responses to the stimuli. Unlike the WM tasks that the author cites, the animals were not taught what to respond to via training and reward. Instead, the animals’ natural responses show that without training they identified the pattern in the sequences (start and end with A), and responded to the non-grammatical sequences that did not follow this pattern.
While I agree with the author that future grammar learning studies could benefit from making greater contact with working memory studies, habituation/dishabituation paradigms also provide a valuable tool to study implicit learning using natural responses. This section should be revised in light of these points.

Minor comments
In the Abstract in the review document I received (but not the version in the paper) “ABnA” is incorrectly written as “ABnB”. This is not an issue in the version of the manuscript that I saw, but this should be checked to make sure the error doesn’t appear in the final article.
I noticed a number of spelling and grammatical errors, listed below. However, this list is not exhaustive and the manuscript should be thoroughly checked before publication.
Table 1: “speaker” spelled as “speker”.
Line 56-57: additional “are” in “…sequences in Test 2 are, however, are lower…”
Line 58: “further” spelled as “furhter”.
Tables 2 and 3: Column heading is “Fraction of trials…”, however proportion would be more accurate for values such as 0.77, 0.60, etc.
Line 60: “$t$-tests”
Table 8: “adjacent” spelled “adjacient”

Reviewer 2 ·

Basic reporting

There are typos and problems with inadequate referencing.

Experimental design

Not applicable

Validity of the findings

Speculation and opinions not always identified as such, and inadequately justified.

Additional comments

Review of Ghirlanda

This short paper reports a reanalysis of the data from a previously published habituation-dishabituation study with squirrel monkeys by Ravignani et al 2013, criticizing some of the statistical methods used in that study. Although more and more journals are requiring open data, which is excellent, this is the first paper that I have reviewed that actually makes use of this, which I think is a useful endeavor. Thus I support publication of this paper in PeerJ, in principle.

In practice, however, this paper has many problems. It fails to separate criticisms of the Ravignani paper from what might be seen as general criticism of the entire field of habituation-dishabituation studies (which have a long history, in both animals and infants, no reference to which is made in the submitted ms.). I think many of the criticisms made apply to this entire field, but only specify this paper as their target. It also fails to separate matters of fact (if you redo the analysis in different ways, the effect is not significant) from matters of opinion (“categorization of data is bad practice”; “operant testing is better than habituation studies”), and the entire final discussion is essentially orthogonal to the actual data presented (namely, that researchers studying pattern learning in animals should cite the author and colleagues’ work more frequently). The paper also contains many errors (e.g. the very first sentence of the abstract reads “conforming to an ABnB grammar” – no! its an ABnA grammar – and many other typos and minor errors).

For all these reasons I recommend a revise and resubmit decision, and I will be glad to look at a revised version that separates matters of fact from opinion and speculation, and that more clearly distinguishes between standard practice in the field of animal playback studies, and any specific critique of Ravignani’s analysis. (see for example a recent short review by ten Cate 2017)

1. As mentioned above, the paper needs to clearly sort out (and largely eliminate) unsubstantiated opinions, and be more clear that those opinions that are left in (which should mainly be in the Discussion) are opinions. The abstract is a case in point stating “the data indicate very poor generalization” – but this is subjective (“very poor” compared to what?) and depends on the assumptions made during data analysis. The last line says the study “may be fruitfully analyzed as an auditory sequence discrimination task” – of course it can and the authors nowhere state otherwise.
2. The revision needs to make more reference to the large existing literature on habituation-dishabituation studies, and make clear which criticisms apply to that (and not just one study in this tradition by Ravignani and colleagues). Since the beginnings of this paradigm in the infant work of Peter Eimas, and its adoption into animal research by Seyfarth & Cheney, habituation-dishabituation studies have produced many important papers. The key virtue of this method is that it can quickly probe the spontaneous behavior of subjects without any training, and this has been particularly useful in field studies with primates but also in the lab for both infants and animals. The author essentially opines that the traditional learning theory approach (e.g. operant tasks) is superior, when in fact it is widely recognized that both paradigms have their strengths and weaknesses and have both made important contributions. This difference is obscured by using the word “training” for the exposure stage (which provides no contingent feedback, and is thus not “training” at all, in the typical sense), so it is better to refer to this as “exposure”.
3. Continuing the point above, the author criticizes the use of a categorization of the monkey data into “look” and “no look” with the brief statement that “Generally dichotomization is not advised as it leads to loss of information”, citing two psychology journals. But this is standard practice in most animal playback experiments, where look/no look is the most basic and simple possible behavioral measure of whether the animals looked or not. And in any case, whether categorization leads to “loss of information” vs. “loss of noise” depends on the data themselves, and the sources of noise. There are thousands of reasons an animal might look away briefly (e.g. to check on a conspecific, or due to a noise or movement) that can’t be controlled, and if this leads to a response being classed as 2 looks vs. 1, it may (and in the case of these data, probably does) simply emphasize these sources of noise. Line 84-85 essentially reinforces this point. The author is entitled to his opinion about categorization but it should be stated as such and justified, and the point above about the value of categorization in the face of noise should be explicitly recognized as well.
4. “Inclusion of Training Sequences” – This whole section needs to be rewritten. Every test stimulus in the Ravignani study was novel: the only thing some test sequences had in common were the PATTERN not the sequences themselves. Since each sequence was played once and there were hundreds of them played, saying that the study used “the same stimuli to demonstrate the grammar” is both factually incorrect (they were all novel) and implies unrealistically that the monkeys could somehow remember individual strings from the exposure phase. The point that the author can validly make is that the monkeys perhaps could not discriminate between all these different strings, which is of course likely, but that is part of the design of the experiment. Furthermore, I know of not a single published playback study which did not play novel stimuli that followed the exposure pattern, making the opinion expressed in line 110-11 (“it would seem prudent to exclude” such sequences) very much an outlier in this field of research. The author should at the very least recognize that this is standard practice, and criticize the field rather than this study.
5. The Discussion is rather disconnected from the rest of the paper. It seems to attribute opinions to Ravignani et al (e.g. that “monkeys can conceptualize grammars”) that they are unlikely to hold. Statements like “monkeys’ performance was unimpressive” are unjustified – the comparison papers given used thousands of training trials with contingent feedback to achieve the high performance cited. Comparing operant training with habituation is comparing apples and oranges. Similarly with “only 6% of the variance” explained (also mentioned in the Abstract) – the only way to evaluate this as good or poor is explicit comparison with studies employing similar methods. Given how much variance in any animal study is caused by extraneous factors beyond experimenter control, this may be a pretty high amount.

In summary, I am sympathetic to the overall aims of this paper, and the critical spirit behind it, but the author has quite a bit of work and rewriting (and probably getting familiar with the habituation literature) to do before the ms. can achieve these goals.



Typos:
1. Abstract ABnB -> ABnA
2. Line 58 “furhter”
3. L 60 “$t$-tests”

Reference: 1. ten Cate C. Assessing the uniqueness of language: Animal grammatical abilities take center stage. Psychonomic Bulletin & Review. 2016;in press.

---

## Round 0.2 · Minor Revisions

· Academic Editor

Minor Revisions

I was very fortunate that both expert reviewers who previously reviewed your MS were available to review the revision. Both seem satisfied that you have addressed the majority of their comments; however, they each have some minor points they'd like you to address/correct before the MS is accepted. I think their comments are more than reasonable and the required revisions are very minor. In addition, please correct the misspelling of analysis on line 73. Please do not place tables in the middle of a paragraph (e.g., Tables 4-6). It seems it would make sense to place figures 6 and 7 right next to each other given the request for the reader to compare them on line 96. Please place a comma after Thus, on line 124. "Or not" is not needed on lines 146-147 and 160, 161. Please delete.

Reviewer 1 ·

Basic reporting

Reporting is clear throughout

Experimental design

N/A

Validity of the findings

Analyses are appropriate and robust, although see my one remaining comment, below.

Additional comments

Re-review of “Can squirrel monkeys learn an ABnA grammar? A re-evaluation of Ravignani et al. (2013)” Ghirlanda, S.

I reviewed a previous version of this manuscript and made a number of comments about issues that needed to be addressed before publication. The author has taken these comments seriously, and I am much happier to recommend the much improved revised manuscript for publication. I only have one remaining comment, which unfortunately may not have been clear in my previous review; I apologize for this.

As the author argues, dichotomization of data brings with it both advantages and disadvantages. In general, I agree with the points raised in the revised manuscript on this topic. However, there is at least one additional point in relation to this study that should at least be discussed. In this experiment, Ravignani et al. counted the number of times a monkey turned its head towards the speaker within 7 seconds of stimulus presentation. Ghirlanda argues that a higher number looking responses might represent more interest in a particular testing sequence, and therefore that this information should not necessarily be discounted by dichotomizing the data. This seems reasonable, however an important counter argument is not presented in the manuscript. If a monkey were particularly interested in a given stimulus, it is possible that they might turn towards the speaker and continue to look in that direction for a longer period of time (resulting in only one response, rather than many). Ideally, the data could be analyzed based on the duration of looking responses (which would capture both trials that produced many, brief responses and few, prolonged responses). However, I assume this data was not collected (as it is difficult to measure exact onsets and offsets of responses) or is otherwise not available. Nevertheless, this point should at least be clearly discussed in the case for and against dichotomizing data – it is possible that a monkey that produces many looking responses is simply scanning the environment (including occasional glances towards the speaker), while a monkey who is particularly interested in a testing sequence might stare fixedly towards the speaker (but only make one looking response). This suggestion actually fits the data presented in Figure 1 quite neatly, for both experiments. Here, when we consider sequences with 0 or 1 response only, we see what appears to be a clear interaction between grammatical and non-grammatical stimuli (the animals are more likely to respond to a non-grammatical stimulus and to not respond to a grammatical one). This pattern disappears when trials producing 2, 3 and 4 responses are considered, consistent with the idea that the animal was simply looking around more freely, regardless of the stimulus that was presented. Of course without access to all of the data, it is difficult to formally test this possibility. However, the author should address this point when discussing the merits of dichotomization in this case.

I apologize that I failed to make this point clearly in my previous review of this article. I am happy with all other aspects of the revision, and would recommend publication if this point could be addressed and discussed.

Reviewer 2 ·

Basic reporting

OK, but see below about adding N

Experimental design

NA

Validity of the findings

NA

Additional comments

Re-review of Ghirlanda

The paper has been substantially rewritten, taking most of the reviewers’ comments into account, and is greatly improved. But I think it still needs some work before publication.

Here are the changes, pretty easy, that I think would be necessary for publication:

1. The current ms. states in the abstract that “generalization over n is required” to show learning of a grammar. This does not characterize the published literature on artificial grammar learning (AGL) well at all (even though I am inclined to agree with the statement). For example both human studies (Perruchet and Rey) and animal studies (Rey & Fagot) have failed to test for generalization.

It would be better to say something like:
“Generalization over n has been argued to be an important criterion for ‘learning’ in artificial grammar learning studies (e.g. by Fitch & Friederici 2012), but has not been tested for or shown in many published studies in this field (e.g. Perruchet & Rey 2005; Rey et al 2012).”

Fitch WT, Friederici AD. 2012. Artificial Grammar Learning Meets Formal Language Theory: An Overview. Philosophical Transactions of The Royal Society B 367:1933-1955.
Perruchet P, Rey A. 2005. Does the mastery of center-embedded linguistic structures distinguish humans from nonhuman primates? Psychonomic Bulletin and Review 12:307-313.
Rey A, Perruchet P, Fagot J. 2012. Centre-embedded structures are a by-product of associative learning and working memory constraints: evidence from baboons (Papio papio). Cognition 123:180-184.

2. The Tables illustrating the statistical results should give the N of stimuli played (this is particularly relevant for Table 5, to interpret the failure to find signficance for some tests (as observed previously by the other reviewer regarding the first ms.).

3. Section 3.3 needs the most work. First, this section should restate what exactly Test 2 represented: “..where the roles of the high and low tones were switched, to determine whether animals would generalize from ABnA to BAnB stimuli”

Also, “rejected confidently” is too strong, as it rests on a statistical test with low power due to low N, and an assumption that “habituation typically proceeds over many trials”. This latter statement may typically be true, but there are many examples of one-trial learning in the literature.

So I think the sentence “This possibility can be rejected confidently” should be deleted, and after summarizing the arguments the first paragraph could simple say “These observations are inconsistent with Ravignani et al’s argument”. And at the very end of the section something like “I conclude that the evidence for generalization from ABnA to BAnB is weak”

4. I don’t feel like the author took my earlier request to say something more about this type of auditory playback experiments seriously. He still only mentions one study in this method (the Ravignani study).

He also misnames this type of test calling it an “auditory habituation task” which would include a wide variety of tasks (e.g. oddball paradigm in ERP, etc) that have nothing to do with the study being discussed.

This is a familiarization/discrimination paradigm (sometimes called a habituation/dishabituation paradigm) and that should be clarified. Also, at least one of two other papers using this method should be cited, eg. Eimas (who invented it, for babies) and Cheney & Seyfarth (who I think first applied it to nonhuman primates):
Eimas PD, Siqueland ER, Jusczyk PW, Vigorito J. 1971. Speech perception in infants. Science 171:303-306.
Cheney DL, Seyfarth RM. 1988. Assessment of meaning and the detection of unreliable signals by vervet monkeys. Anim Behav 36:477-486.
This paper gives a brief review:
Fischer J. 2006. Categorical perception in animals. In: Brown K, editor. Encyclopedia of Language & Linguistics – Second Edition  Oxford, UK: Elsevier p 248-251.

5. Finally the logic of Section 3.5 (“Contribution of stimulus generalization” remains unclear to me. The whole idea of the Ravignani study (like most studies in this paradigm) is that monkeys should generalize over the A and B categories. So the statement that “These experiences may have been sufficient for stimulus generalization to influence monkey behaviour” is simply stating the obvious and should be revised to state clearly whatever it is the author is getting at here.

---

## Round 0.3 · accepted · Accept

· Academic Editor

Accept

Thank you for your careful attention to the last round of reviews. I am inclined to agree with Reviewer 1's point about the duration of viewing versus the number of orientations and would have preferred if you had addressed this as a possibility for future research in your concluding statements, but I am willing to concede on your argument that your comment need not address all issues with the original study. Thank you for submitting such thoughtful work to PeerJ.